# AsyncDiff: Parallelizing Diffusion Models by Asynchronous Denoising

**Zigeng Chen, Xinyin Ma, Gongfan Fang, Zhenxiong Tan, Xinchao Wang**[*]
National University of Singapore
zigeng99@u.nus.edu, xinchao@nus.edu.sg

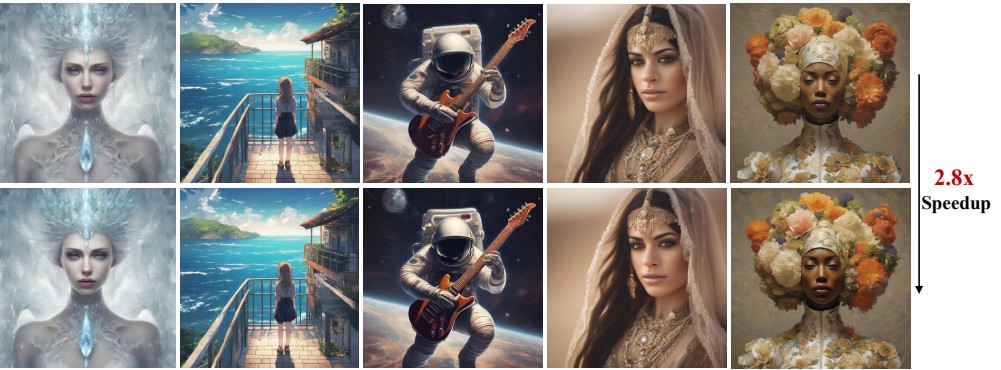

Figure 1: We introduce a new distributed acceleration paradigm that attains a 2.8x speed-up on Stable Diffusion XL while maintaining **pixel-level consistency**, using four NVIDIA A5000 GPUs.

## Abstract

Diffusion models have garnered significant interest from the community for their great generative ability across various applications. However, their typical multi-step sequential-denoising nature gives rise to high cumulative latency, thereby precluding the possibilities of parallel computation. To address this, we introduce *AsyncDiff*, a universal and plug-and-play acceleration scheme that enables model parallelism across multiple devices. Our approach divides the cumbersome noise prediction model into multiple components, assigning each to a different device. To break the dependency chain between these components, it transforms the conventional sequential denoising into an asynchronous process by exploiting the high similarity between hidden states in consecutive diffusion steps. Consequently, each component is facilitated to compute in parallel on separate devices. The proposed strategy significantly reduces inference latency while minimally impacting the generative quality. Specifically, for the Stable Diffusion v2.1, *AsyncDiff* achieves a 2.7x speedup with negligible degradation and a 4.0x speedup with only a slight reduction of 0.38 in CLIP Score, on four NVIDIA A5000 GPUs. Our experiments also demonstrate *AsyncDiff* can be readily applied to video diffusion models with encouraging performances. Code is available at https://github.com/czg1225/AsyncDiff

## 1 Introduction

Diffusion models [13] stand out in generative modeling and have significantly advanced various fields including text-to-image [43, 41, 45, 46, 72, 78] and text-to-video generation [64, 9, 61, 21, 2],

---

[*]Correspoding Author

38th Conference on Neural Information Processing Systems (NeurIPS 2024).

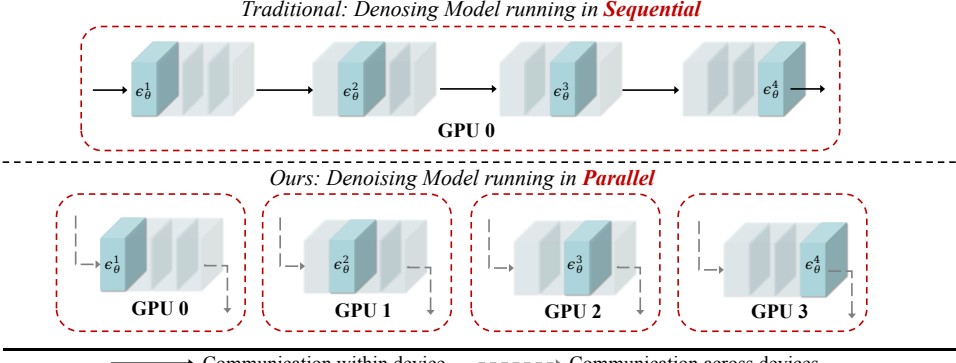

Figure 2: By preparing each component's input beforehand, we enable parallel computation of the denoising model, which substantially reduces latency while minimally affecting quality.

image translation [49, 56, 23], audio generation[22, 14, 44], style transfer[62, 4, 17], low-level vision tasks [47, 60, 40, 26, 8, 70, 3], image editing [19, 66, 51, 77], and 3D model generation [42, 18, 37], among others. However, their widespread application is hindered by the high latency inherent in their multi-step sequential denoising process. This issue becomes more pronounced as the complexity and size of the models increase to enhance generative quality.

In response to these challenges, significant research efforts are directed toward enhancing the efficiency of diffusion models. Notably, training-free acceleration methods have garnered increasing popularity due to their low cost and convenience. Numerous studies [35, 63, 76, 67, 53, 25, 33, 57, 34] improve inference speed by skipping redundant calculations in the denoising process. As computational resources grow rapidly, distributing computations across multiple devices has become a more promising approach. Recent advances [52, 24, 58] demonstrate that using distributed computing to parallelize inference effectively increases the acceleration ratio for diffusion models while maintaining acceptable generative quality. Though these methods succeed in parallelizing the diffusion models, they require iterative refining [52] or displaced patch parallelism [24], resulting in a larger number of model evaluations or low GPU utilization correspondingly.

Thus, we wish to propose a new parallel paradigm for diffusion, akin to the model parallelism in distributed computing [15, 38, 28, 16, 39, 65], which divides the denoising model into several components to be distributed on different GPUs. The primary challenge lies in the inherent sequential denoising process of diffusion models. Each step in this process depends on the completion of its predecessor, forming a dependency chain that impedes parallelization and significantly increases inference latency. Our approach seeks to disrupt this chain, allowing for the parallel execution of the denoising model while closely approximating the results of the sequential process.

In this paper, we introduce *AsyncDiff*, a universal, distributed acceleration paradigm that innovatively explores model parallelism in diffusion models. As shown in Fig 2, our method sequentially partitions the heavyweight denoising model $\epsilon_\theta$ into multiple components $\{\epsilon_\theta^n\}_{n=1}^N$ based on computational load, assigning each to a separate device. Our core idea lies in decoupling the dependencies between these cascaded components by leveraging the high similarity in hidden states across consecutive diffusion steps. After the initial warm-up steps, each component takes the output from the previous component's prior step as the approximation of its original input. This transforms the traditional sequential denoising into an asynchronous process, allowing components to predict noise for different time steps in parallel. Additionally, we incorporate stride denoising to skip redundant calculations and reduce the frequency of communication between devices, further enhancing efficiency.

Through extensive testing across multiple base models, our method effectively distributes the computational burden across various devices, substantially boosting inference speed while maintaining quality. Specifically, with the text-to-image model Stable Diffusion v2.1 [43], our method achieves a 1.8x speedup with only a marginal 0.01 drop in CLIP Score [11], and a 4.0x speedup with a slight 0.38 reduction in CLIP Score on two and four NVIDIA A5000 GPUs, respectively. For video diffusion models, AnimateDiff [9] and Stable Video Diffusion [2], our approach significantly reduces latency by tens of seconds, effectively preserving video quality.

In summary, we present a novel distributed acceleration method for diffusion models that significantly reduces inference latency with minimal impact on generation quality. This is achieved by replacing the sequential denoising process with an asynchronous process, allowing each component of the denoising model to run independently across different devices. Extensive experiments on both image and video diffusion models strongly demonstrate the effectiveness and versatility of our method.

## 2 Related Works

**Diffusion Models**. Diffusion models have attracted significant attention due to their powerful generative capabilities across various tasks. Sohl-Dickstein et al. [54] first proposed diffusion probabilistic models. Ho et al. [13] with the introduction of Denoising Diffusion Probabilistic Models (DDPM), enhancing training efficiency and generation quality. Rombach et al. [43] advanced these models by incorporating latent spaces, enabling high-resolution image generation. Despite these advancements, the high latency of the iterative denoising process remains a limitation.

**Inference Acceleration**. Training-based acceleration methods focus on reducing sampling steps [48, 71, 32, 50, 69] or optimizing model architectures [27, 80, 7, 73, 68, 6]. However, these methods incur high training costs and complexity. Training-free methods are gaining popularity due to their ease of use. Some approaches develop fast solvers for SDE or ODE to improve sampling efficiency [31, 1, 30, 74, 81]. Other works [35, 63, 76, 67, 53, 25, 33, 79] observed special characteristics of diffusion models and skipped the redundant computation within the denoising process.

**Parallelism**. The parallelism strategy presents a promising yet underexplored approach to accelerating diffusion models. ParaDiGMS [52] implements Picard iterations for parallel sampling, yet its practical speed-up ratio is modest, and it struggles to maintain consistency with original outputs. Faster Diffusion [25] introduces encoder propagation but significantly compromises quality, and its parallelization remains theoretical. Distrifusion [24] adopts patch parallelism, dividing high-resolution images into sub-patches to facilitate parallel inference on each patch by reusing stale activation maps from each layer. However, this approach lacks flexibility across different data types or tasks, often encountering low resource utilization. Furthermore, its reliance on reusing per-layer activation maps greatly increases GPU memory demands thus introducing additional challenges for realistic applications. In contrast, our method uniquely implements model parallelism through asynchronous denoising, achieving substantial acceleration while maintaining a stable resource usage ratio and minimal impact on quality.

## 3 Methods

### 3.1 Preliminary

Diffusion models [13] are a dominant class of generative models that transform Gaussian noise into complex data distributions via a Markov process. The forward process is defined by:

$$q(x_t|x_{t-1}) = \mathcal{N}(x_t; \sqrt{1 - \beta_t}x_{t-1}, \beta_t I), \tag{1}$$

where $\{\beta_t\}$ progressively increases noise until the data becomes indistinguishable from noise. The reverse process, essential for data reconstruction, involves iterative denoising:

$$p_\theta(x_{t-1}|x_t) = \mathcal{N}(x_{t-1}; \mu_\theta(x_t, t), \sigma_t^2 I), \tag{2}$$

where $\mu_\theta(x_t, t)$ is the predicted mean and $\sigma_t^2$ is the variance. For DDIMs [55], the reverse update is deterministic:

$$x_{t-1} = \sqrt{\frac{\alpha_{t-1}}{\alpha_t}}x_t + \sqrt{1 - \alpha_{t-1}}\left(1 - \sqrt{\frac{1 - \alpha_t}{\alpha_{t-1}}}\right)\epsilon_\theta(x_t, t), \tag{3}$$

where $\alpha_t$ is the cumulative product of $(1 - \beta_t)$. These processes are computationally intensive, influencing the quality of generated samples and necessitating efficient inference methods for practical applications.

### 3.2 Asynchronous Diffusion Model

Traditional diffusion models employ a sequential and synchronous denoising process. At each time step $t$, the noise-prediction model $\epsilon_\theta$ estimates the noise $\epsilon_t$ based on the noisy image $x_t$ and the time

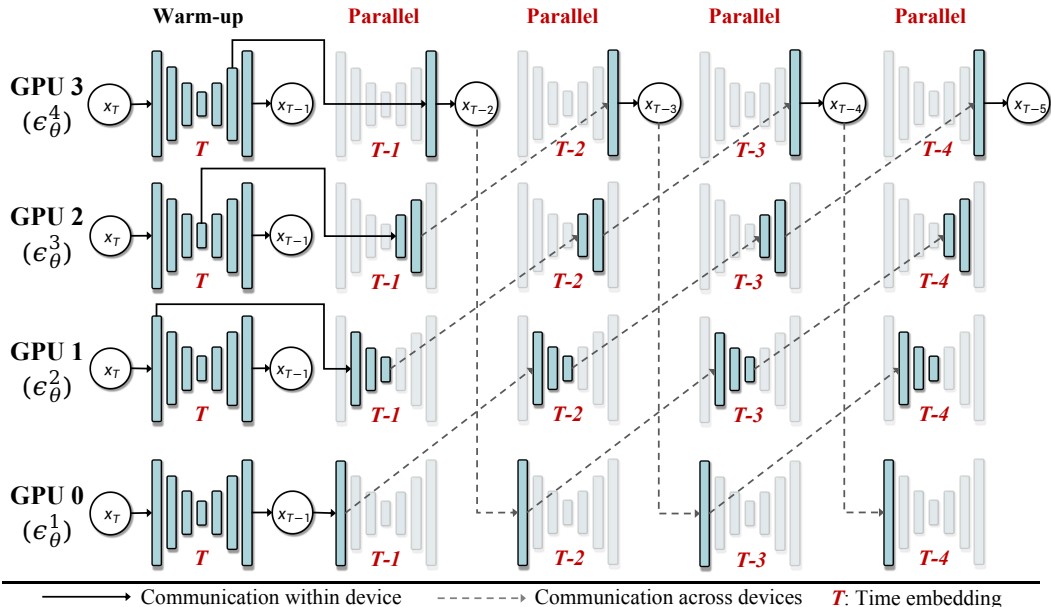

Figure 3: Overview of the asynchronous denoising process. The denoising model $\epsilon_\theta$ is divided into four components $\{\epsilon_\theta^n\}_{n=1}^4$ for clarity. Following the warm-up stage, each component's input is prepared in advance, breaking the dependency chain and facilitating parallel processing.

embedding $t$. The image for the next step, $x_{t-1}$, is then generated using a sampler function $S(x_t, \epsilon_t, t)$. This process is iterative, where the generation of $\epsilon_t$ at each step is dependent on the completion of the previous denoising step, making the process slow, particularly when $\epsilon_\theta$ is computationally intensive.

To address the limitations of high latency in diffusion models, leveraging multiple GPUs for distributed inference is a promising solution. Existing studies primarily focus on patch parallelism [24], where the input image is divided into patches, each processed on a different GPU. While this strategy efficiently distributes computational loads, it still retains the bottleneck of sequential denoising, as each patch must undergo the complete denoising process iteratively. In contrast, our asynchronous diffusion model innovatively introduces a model parallelism strategy. By approximating the sequential denoising as an asynchronous process, this approach enables parallel inference of the noise prediction model, effectively reducing latency and breaking the constraints of sequential execution.

**Asynchronous Denoising**. Figure 3 illustrates our approach to the asynchronous denoising. For a denoising process consisting of $T$ steps, the initial $w$ steps are designated as a warm-up phase, where $w$ is significantly smaller than $T$. During this phase, the denoising model $\epsilon_\theta$ operates using standard sequential inference. After warm-up steps, rather than splitting the input image, we partition the denoising model $\epsilon_\theta$ into $N$ sequential components, expressed as $\epsilon_\theta = \{\epsilon_\theta^1, \epsilon_\theta^2, ..., \epsilon_\theta^N\}$. Each component is divided to handle a comparable computational load and assigned to a distinct device. This equitable division aims to equalize the time cost of each component to approximately $l(\epsilon_\theta)/N$, thus minimizing the overall maximum latency. In this setup, original noise prediction for $x_t$ can be represented as a cascading operation through these sub-models, defined mathematically as:

$$\epsilon_t = \epsilon_\theta(x_t, t) = \epsilon_\theta^N(\epsilon_\theta^{N-1}(\ldots \epsilon_\theta^2(\epsilon_\theta^1(x_t, t), t) \ldots, t), t). \tag{4}$$

Although each device can independently compute its assigned component, the dependency chain persists because the input for each component $\epsilon_{\theta,n}$ is derived from the output from its preceding component $\epsilon_{\theta,n-1}$. Therefore, despite the distribution of model components across multiple devices, full parallelization is constrained by these sequential dependencies.

Our principal innovation is to break the dependency between cascaded components by utilizing hidden features from previous steps. Observations indicate that the hidden states of each block in the denoising model always exhibit substantial similarity across adjacent time steps. Leveraging this, each component at time step $t$ can take the output from the preceding component at time step $t-1$ as the approximation of its original input. Specifically, the $n$-th component $\epsilon_\theta^n(, t)$ receives the output of

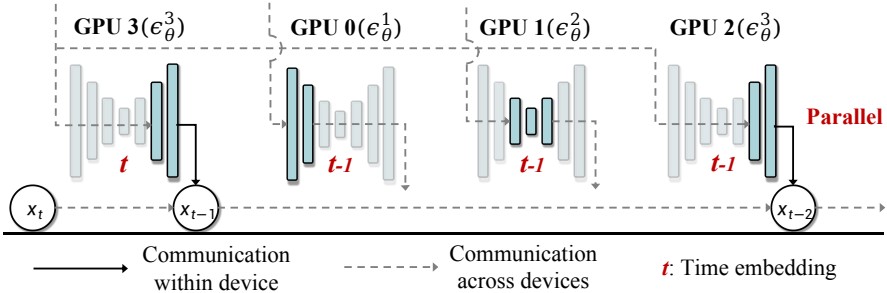

Figure 4: Illustration of stride denoising. The model $\epsilon_\theta$ is divided into three components $\{\epsilon_\theta^n\}_{n=1}^3$, with a stride $S$ of 2 for clarity. Components $\epsilon_\theta^1$ and $\epsilon_\theta^2$ are skipped at time step $t$. A single parallel batch results in the completion of denoising for two steps, producing $x_{t-1}$ and $x_{t-2}$.

$\epsilon_\theta^{n-1}(\cdot, t-1)$. This alteration allows the noise prediction for $x_t$ to be represented as follows:

$$\epsilon_t = \epsilon_\theta^N(\epsilon_\theta^{N-1}(\ldots \epsilon_\theta^2(\epsilon_\theta^1(x_{t+N-1}, t+N-1), t+N-2)\ldots, t+1), t). \tag{5}$$

In this new framework, noise prediction $\epsilon_t$ is derived from components executed across $N$ previous time steps. This transforms the denoising process from sequential to asynchronous, as the prediction of noise $\epsilon_t$ already begins before denoising at step $t+1$ is completed. At each time step, the $N$ components are running as parts of the noise prediction model for the next $N$ steps. Specifically, the $n$-th component $\epsilon_\theta^n$, computed in parallel at time $t$, contributes to the noise prediction for the future time step $t-N+n$. Figure 3 depicts this asynchronous process using a U-net model with $N$ set to 4. The strong resemblance of hidden states between consecutive diffusion steps enables the asynchronous process to closely mimic the denoising results of the original sequential process.

**Model Parallelism**. By transitioning to an asynchronous denoising strategy, the dependencies among components within the same time step are eliminated. This adjustment allows each component's input for time step $t$ to be prepared in advance, enabling the $N$ split components to be processed concurrently across multiple devices. Once computed, the outputs from each component must be stored and then broadcasted to other devices to facilitate parallel processing for subsequent time steps. In contrast, in the traditional sequential denoising process, the time cost for each step accumulates as follows:

$$C_{seq}(t) = C(\epsilon_\theta^1) + C(\epsilon_\theta^2) + \ldots + C(\epsilon_\theta^N). \tag{6}$$

By adopting asynchronous denoising to enable parallel computation of each component, the cost for each time step is now given by:

$$C_{asy}(t) = \max(C(\epsilon_\theta^1), C(\epsilon_\theta^2), ..., C(\epsilon_\theta^N)) + C(\text{comm.}), \tag{7}$$

where $\max()$ represents taking the maximum value, and $C(\text{comm.})$ indicates the communication cost across multiple GPUs. As the model components are equally divided by computational load, their time costs are similar, allowing us to approximate the overall cost of each time step as:

$$C_{asy}(t) \approx \frac{C_{seq}(t)}{N} + C(\text{comm.}). \tag{8}$$

Since the communication overhead $C(\text{comm.})$ is generally much lower than the model's execution time, it leads to significant overall cost reductions. Moreover, increasing $N$ further reduces time costs but complicates the accurate approximation of the original denoising process.

**Stride Denoising**. While asynchronous denoising reduces latency by parallelizing the denoising model, it completes only one denoising step at a time. To enhance efficiency, we introduce stride denoising, which completes multiple denoising steps simultaneously through a single parallel computation. The diagram is illustrated in Figure 4, where we set the stride to 2 for clarity. Unlike the continuous broadcasting of hidden states at each time step, stride denoising broadcasts them every two steps. As depicted, at time step $t$, we conduct denoising alone, and at time step $t-1$, we compute and broadcast the hidden states for the next parallel computation round. Consequently, the hidden states from time step $t$ are not required, allowing us to skip the calculations for $\epsilon_\theta^1$ and $\epsilon_\theta^2$ at this step. In this stride, only $\epsilon_\theta^3(\cdot, t)$, $\epsilon_\theta^1(\cdot, t-1)$, $\epsilon_\theta^2(\cdot, t-1)$, and $\epsilon_\theta^3(\cdot, t-1)$ need computing, all receiving the

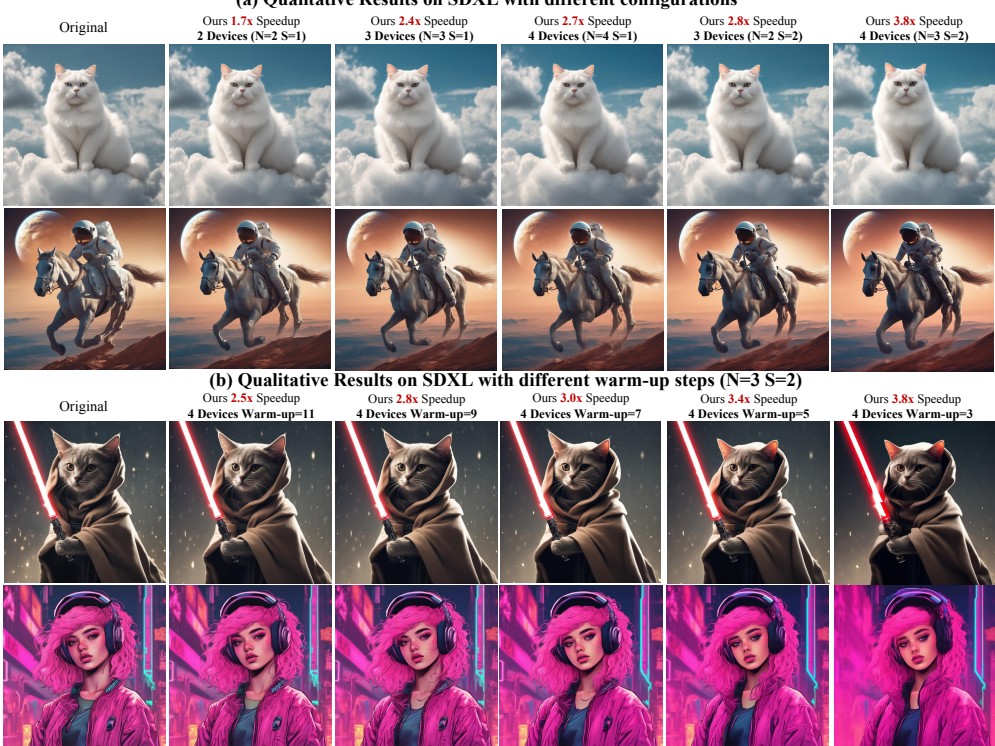

Figure 5: Qualitative Results. (a) Our method significantly accelerates the denoising process with minimal impact on generative quality. (b) Increasing warm-up steps achieves pixel-level consistency with the original output while maintaining a high speed-up ratio.

previously broadcast hidden states, enabling their parallel processing. Both $\epsilon_\theta^3(\cdot, t)$ and $\epsilon_\theta^3(\cdot, t-1)$ share the same feature from $\epsilon_\theta^2(\cdot, t+1)$, so the stride should be kept small to maintain quality. Stride denoising effectively reduces both computational load and communication demands by decreasing the parallel computing rounds needed to complete the process. Compared to the significant improvements it brings in efficiency, the quality sacrifice is minimal and can be entirely compensated for by slightly increasing the warm-up steps. We also illustrate the full schematic of it in Appendix Figure 7.

**Multi-Device Communication**. Parallel inference of the model necessitates efficient communication between devices, as each component $\epsilon_\theta^n$ must access the cached hidden state from the preceding component $\epsilon_\theta^{n-1}$, which resides on a different device. Post each parallel computation batch, each device stores the current hidden state needed for the next parallel batch. These states, encompassing all component outputs, are then broadcast to all participating devices before the next parallel computation batch. Although each component $\epsilon_\theta^n$ primarily uses the cached output of $\epsilon_\theta^{n-1}$ for its input, it may require residual features [10] from other components. Therefore, it's crucial to broadcast the stored states from every component across all devices before each round of parallel computation.

## 4 Experiments

### 4.1 Implementation Details

**Base models**. We validated the broad applicability of *AsyncDiff* through extensive testing on several diffusion models. For text-to-image tasks, we experimented with three versions of Stable Diffusion: SD 1.5, SD 2.1 [43], and Stable Diffusion XL (SDXL) [41]. Additionally, we explored the effectiveness of *AsyncDiff* on video diffusion models using Stable Video Diffusion (SVD) [2] and AnimateDiff [9]. All models were evaluated using 50 DDIM steps. We facilitated communication across multiple GPUs using the *broadcast* operation from *torch.distributed*, powered by the *NVIDIA Collective Communication Library* (NCCL) backend.

Table 1: Quantitative evaluations of *AsyncDiff* on three text-to-image diffusion models, showcasing various configurations. 'N' indicates the number of components into which the model is divided, and 'S' represents the denoising stride. *MACs* quantifies the computational load per device for generating a single image throughout the denoising process.

| Base Model | Configuration | Devices | MACs↓ | latency↓ | Speed up↑ | CLIP Score↑ | FID↓ | LPIPS↓ |
|---|---|---|---|---|---|---|---|---|
| SD 2.1 (Text-to-Image) | Original Model | 1 | 76T | 5.51s | 1.0x | 31.60 | 27.89 | – |
| | + **Ours** (N=2 S=1) | 2 | 38T | 3.03s | 1.8x | 31.59 | 27.79 | 0.2121 |
| | + **Ours** (N=3 S=1) | 3 | 25T | 2.41s | 2.3x | 31.56 | 28.00 | 0.2755 |
| | + **Ours** (N=4 S=1) | 4 | 19T | 2.10s | 2.6x | 31.40 | 28.28 | 0.3132 |
| | + **Ours** (N=2 S=2) | 3 | 19T | 1.82s | 3.0x | 31.43 | 28.55 | 0.3458 |
| | + **Ours** (N=3 S=2) | 4 | 13T | 1.35s | 4.0x | 31.22 | 29.41 | 0.3778 |
| SD 1.5 (Text-to-Image) | Original Model | 1 | 34T | 2.70s | 1.0x | 30.63 | 29.96 | – |
| | + **Ours** (N=2 S=1) | 2 | 17T | 1.52s | 1.8x | 30.62 | 29.94 | 0.1988 |
| | + **Ours** (N=3 S=1) | 3 | 11T | 1.23s | 2.2x | 30.58 | 29.87 | 0.2645 |
| | + **Ours** (N=4 S=1) | 4 | 9T | 1.01 | 2.6x | 30.52 | 30.10 | 0.3073 |
| | + **Ours** (N=2 S=2) | 3 | 9T | 0.94s | 2.9x | 30.46 | 30.98 | 0.3232 |
| | + **Ours** (N=3 S=2) | 4 | 6T | 0.72s | 3.7x | 30.17 | 30.89 | 0.3811 |
| SDXL (Text-to-Image) | Original Model | 1 | 299T | 13.81s | 1.0x | 32.33 | 27.43 | – |
| | + **Ours** (N=2 S=1) | 2 | 150T | 8.00s | 1.7x | 32.21 | 27.79 | 0.2509 |
| | + **Ours** (N=3 S=1) | 3 | 100T | 5.84s | 2.4x | 32.05 | 28.03 | 0.2940 |
| | + **Ours** (N=4 S=1) | 4 | 75T | 5.12s | 2.7x | 31.90 | 29.12 | 0.3157 |
| | + **Ours** (N=2 S=2) | 3 | 75T | 4.91s | 2.8x | 31.70 | 28.99 | 0.3209 |
| | + **Ours** (N=3 S=2) | 4 | 49T | 3.65s | 3.8x | 31.40 | 30.27 | 0.3556 |

Table 2: Quantitative evaluations of the effect of increasing warm-up steps. More warm-up steps can achieve pixel-level consistency with the original output while slightly reducing processing speed.

| Configuration | SD 2.1 | | | SD 1.5 | | | SDXL | | |
|---|---|---|---|---|---|---|---|---|---|
| | Speedup↑ | CLIP↑ | LPIPS↓ | Speedup↑ | CLIP↑ | LPIPS↓ | Speedup↑ | CLIP↑ | LPIPS↓ |
| Original Model | 1.0x | 31.60 | – | 1.0x | 30.63 | – | 1.0x | 32.33 | – |
| Warm-up = 3 | 3.5x | 31.26 | 0.3289 | 3.3x | 30.16 | 0.3676 | 3.8x | 31.40 | 0.3556 |
| Warm-up = 5 | 3.1x | 31.27 | 0.2769 | 3.0x | 30.14 | 0.3304 | 3.4x | 31.60 | 0.2993 |
| Warm-up = 7 | 2.9x | 31.32 | 0.2309 | 2.7x | 30.10 | 0.2839 | 3.0x | 31.77 | 0.2521 |
| Warm-up = 9 | 2.7x | 31.40 | 0.1940 | 2.5x | 30.17 | 0.2354 | 2.8x | 31.92 | 0.2095 |
| Warm-up = 11 | 2.4x | 31.45 | 0.1628 | 2.4x | 30.22 | 0.1927 | 2.5x | 32.01 | 0.1740 |

**Dataset and Evaluation Metrics**. We assess the zero-shot generation capability using the MS-COCO 2017 [29] validation set, which comprises 5,000 images and captions. For image generation, quality is measured by the CLIP Score (on ViT-g/14) [11] and Fréchet Inception Distance (FID) [12], with LPIPS [75] used to check consistency with original outputs. In video generation, quality is evaluated by averaging the CLIP Score across all frames of a video. We also report MACs per device and latency to gauge efficiency comprehensively. All latency measurements were conducted on NVIDIA A5000 GPUs equipped with NVLINK Bridge.

## 4.2 Experimental Results on Image Diffusion Models

**Improvements on Base Models**. Table 1 displays our acceleration outcomes for three fundamental image diffusion models under various configurations. In this context, 'N' represents the number of segments into which the denoising model is divided, and 'S' denotes the stride of denoising for each parallel computation batch. Our approach, *AsyncDiff*, not only significantly accelerates processing but also minimally impacts generative quality. The speedup ratio is almost proportional to the number of devices used, demonstrating efficient resource utilization. Visualization results in Figure 5 (a) illustrate the high generative quality achieved even with substantially reduced latency. Although achieving pixel-level consistency with the original output is challenging at high acceleration ratios, the generated image still effectively conveys the semantic information in the prompt, which is crucial for generative results.

**Pixel-level Consistency by Warm-up.** In Table 2, we explore the balance between pixel-level consistency and processing speed by adjusting the warm-up steps in the diffusion models. As the initial steps of these models play a crucial role in reconstructing the global structure based on text prompts [76], a modest increase in warm-up steps can significantly enhance consistency with the

Table 3: Quantitative comparison with other parallel acceleration methods. To ensure a fair comparison with Distrifusion, we increased the warm-up steps in our method to match the speedup ratio of Distrifusion, allowing us to fairly compare generation quality and resource costs.

| Method | Speed up↑ | Devices | MACs↓ | Memory↓ | CLIP Score↑ | FID↓ | LPIPS↓ |
|---|---|---|---|---|---|---|---|
| Original Model | 1.0x | 1 | 76T | 5240MB | 31.60 | 27.87 | – |
| Faster Diffusion | 1.6x | 1 | 57T | 9692MB | 30.84 | 29.95 | 0.3477 |
| Distrifusion | 1.6x | **2** | **38T** | 6538MB | **31.59** | 27.89 | **0.0178** |
| Ours (N=2 S=1) | 1.6x | **2** | 44T | **5450MB** | **31.59** | **27.79** | 0.0944 |
| Distrifusion | 2.3x | 4 | **19T** | 7086MB | 31.43 | 27.97 | 0.2710 |
| Ours (N=2 S=2) | 2.3x | **3** | 20T | **5516MB** | **31.49** | **27.71** | **0.2117** |
| Distrifusion | 2.7x | 8 | **10T** | 7280MB | 31.31 | 28.12 | 0.2934 |
| Ours (N=3 S=2) | 2.7x | **4** | 14T | **5580MB** | **31.40** | **28.03** | **0.1940** |

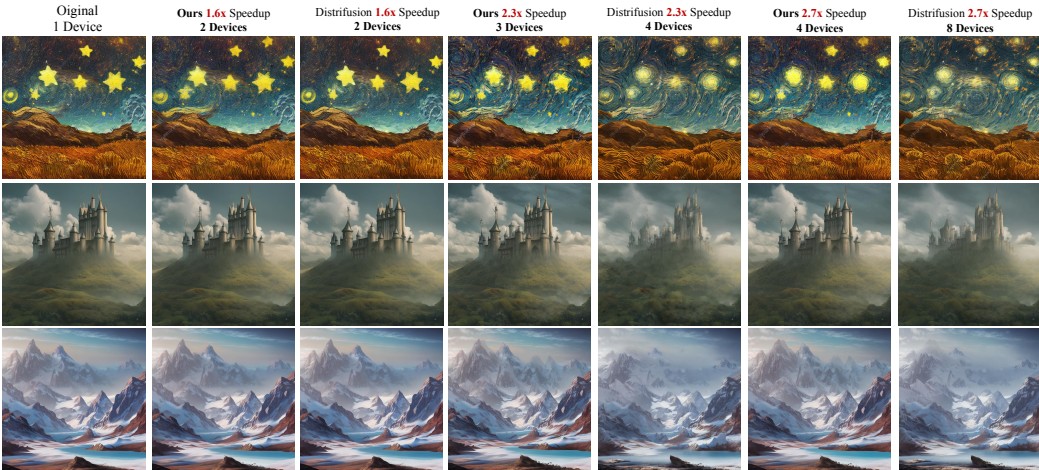

Figure 6: Qualitative Comparison with Distrifusion on SD2.1. At the same acceleration ratio, *AsyncDiff* outperforms in generating higher quality and more consistent images with the original.

original images. Figure 5(b) illustrates this trend with qualitative comparisons of generative results on SDXL using gradually increasing warm-up steps. Increasing the warm-up steps to 9 achieves visual indistinguishability from the original output while maintaining an impressive 2.8x acceleration ratio.

**Comparison with Acceleration Baselines**. We evaluated our *AsyncDiff* method on SD 2.1 against two other parallel acceleration methods: Faster Diffusion [25] and Distrifusion [24]. Faster Diffusion employs encoder propagation but compromises significantly on generative quality. As its parallelism maintains theoretical and lacks a multi-device implementation, we cannot measure its realistic latency with more than one GPU. Its ideal speed-up on 2 devices is about 1.9x. Distrifusion, on the other hand, uses patch parallelism for distributed acceleration but faces potential issues with low resource utilization and high GPU memory demands.

According to Table 3, our method achieves the same operational speed using only 4 GPUs and 3 GPUs as Distrifusion does with 8 GPUs and 4 GPUs, respectively. Additionally, our method requires almost the same amount of memory as the original setup, whereas Distrifusion significantly increases memory requirements, posing extra challenges for practical applications. In terms of generative quality, *AsyncDiff* and Distrifusion both mirror the original diffusion model's performance at a 1.6x acceleration ratio. However, at higher speedup ratios of 2.3x and 2.7x, our method demonstrates significantly superior generative quality. Qualitative comparisons in Fig 6 further show that *AsyncDiff* maintains better pixel-level consistency with the original input compared to Distrifusion.

### 4.3 Experimental Results on Video Diffusion Models

As presented in Table 4, we conducted experiments with different configurations on two video diffusion models: SVD [2] (25 frames), and AnimentDiff [9] (16 frames), to demonstrate the efficacy

Table 4: Quantitative evaluations of *AsyncDiff* on text-to-video and image-to-video diffusion models. We present the results with various configurations.

| Base Model | Configuration | Devices | MACs↓ | latency↓ | Speed up↑ | CLIP Score↑ |
|---|---|---|---|---|---|---|
| AnimateDiff (Text-to-Video) | Original Model | 1 | 786T | 43.5s | 1.0x | 30.65 |
| | **+ Ours** (N=2 S=1) | 2 | 393T | 24.5s | 1.8x | 30.65 |
| | **+ Ours** (N=3 S=1) | 3 | 262T | 19.1s | 2.3x | 30.54 |
| | **+ Ours** (N=2 S=2) | 3 | 197T | 14.2s | 3.0x | 30.32 |
| | **+ Ours** (N=3 S=2) | 4 | 131T | 11.5s | 3.8x | 30.20 |
| SVD (Image-to-Video) | Original Model | 1 | 3221T | 184s | 1.0x | 26.88 |
| | **+ Ours** (N=2 S=1) | 2 | 1611T | 101s | 1.8x | 26.66 |
| | **+ Ours** (N=3 S=1) | 3 | 1074T | 80s | 2.3x | 26.56 |
| | **+ Ours** (N=4 S=1) | 4 | 805T | 68s | 2.7x | 26.19 |

Table 5: Effect of stride denoising on SD 2.1. Stride denoising significantly lowers overall latency and the communication cost while only slightly compromising the generative quality

| Configuration | MACs↓ | Latency↓ | Speedup↑ | Communication | | CLIP Score↑ |
|---|---|---|---|---|---|---|
| | | | | Nums↓ | Latency↓ | |
| *AsyncDiff* (3 devices) w/o stride denoising | 25T | 2.41s | 2.3x Faster | 49 times | 0.23s(9.5%) | **31.56** |
| *AsyncDiff* (3 devices) w/ stride denoising | **19T** | **1.82s** | **3.0x Faster** | **25 times** | **0.12s(6.6%)** | 31.43 |
| *AsyncDiff* (4 devices) w/o stride denoising | 19T | 2.10s | 2.6x Faster | 49 times | 0.40s(19.0%) | **31.40** |
| *AsyncDiff* (4 devices) w/ stride denoising | **13T** | **1.35s** | **4.0x Faster** | **25 times** | **0.10s(7.4%)** | 31.22 |

of our method. Video generation, often constrained by exceptionally high latency and substantial computation load, greatly benefits from our approach. For a 50-step video diffusion model, *AsyncDiff* significantly reduces latency—by tens or even hundreds of seconds—while preserving the quality of generated content. Qualitative results shown in the Appendix. D further corroborate the effectiveness of our method. *AsyncDiff* achieves an impressive acceleration ratio of over three times while still producing videos that closely match the prompt descriptions, ensuring the rationality of actions and details. These findings highlight the substantial potential of *AsyncDiff* in accelerating the inference process of video diffusion models.

### 4.4 Effect of Stride Denoising

We introduce stride denoising to further enhance the efficiency of the asynchronous denoising process. Stride denoising completes multiple steps simultaneously through a single parallel computation, reducing the number of parallel rounds and communication frequency across devices. For a diffusion process with $T$ steps and warm-up step $W$, the number of broadcasts decreases from $T - W$ to $(T - W)//2$ with a stride of 2. This strategy also reduces the computational load on each device by skipping unnecessary calculations. Table 5 shows the effects of stride denoising in our parallel framework with 3 and 4 devices. Stride denoising significantly lowers overall latency and the proportion of communication time, especially as the number of devices used increases. While stride denoising slightly impacts generation quality, this effect is minimal and can be mitigated by a modest increase in warm-up steps, preserving efficiency and maintaining quality.

### 4.5 Compatibility with Various Samplers

With the recent rise of advanced sampling algorithms for diffusion models, a key concern is whether the acceleration method can adapt to various samplers. AsyncDiff is a universal method that can be combined with different samplers, such as the DDIM sampler [55] and DPM-Solver [31]. In Table 7, we present the quantitative evaluation of AsyncDiff on SD 2.1 using the DDIM sampler. Compared to using fewer DDIM steps, our method achieves significantly better generation quality at similar speeds, with the improvement becoming more pronounced as speedup increases. Table 6 presents the quantitative evaluation of AsyncDiff on SD 2.1 with the DPM-Solver sampler. At the same speedup ratio, AsyncDiff significantly enhances generation quality compared to the baseline. Qualitative results are also provided in the Appendix figures, demonstrating that our method achieves considerable acceleration while maintaining high consistency with the original output.

Table 6: Quantitative evaluations of AsyncDiff using DPM-Solver sampler on SD 2.1

| Method | Speed up ↑ | MACs ↓ | CLIP Score ↑ | FID ↓ |
|---|---|---|---|---|
| DPM-Solver 25steps | 1.0x | 76T | 31.57 | 28.37 |
| DPM-Solver 15steps | 1.6x | 46T | 31.52 | 28.89 |
| **Ours (N=2 S=1)** | 1.6x | **38T** | **31.58** | **27.71** |
| DPM-Solver 10steps | 2.2x | 30T | 31.29 | 29.28 |
| **Ours (N=3 S=1)** | 2.2x | **25T** | **31.36** | **28.20** |

Table 7: Quantitative evaluations of AsyncDiff using DDIM sampler on SD 2.1

| Method | Speed up ↑ | MACs ↓ | CLIP Score ↑ | FID ↓ |
|---|---|---|---|---|
| Original | 1.0x | 76T | 31.60 | 27.89 |
| DDIM 27steps | 1.8x | 41T | 31.53 | 28.43 |
| **Our AsyncDiff (N=2 S=1)** | 1.8x | **38T** | **31.59** | **27.79** |
| DDIM 21steps | 2.3x | 32T | 31.46 | 29.09 |
| **Our AsyncDiff (N=3 S=1)** | 2.3x | **25T** | **31.56** | **28.00** |
| DDIM 15steps | 3.0x | 23T | 31.26 | 30.12 |
| **Our AsyncDiff (N=2 S=2)** | 3.0x | **19T** | **31.43** | **28.55** |
| DDIM 11steps | 4.0x | 17T | 30.99 | 32.25 |
| **Our AsyncDiff (N=3 S=2)** | 4.0x | **13T** | **31.22** | **29.41** |

Table 8: Acceleration Ratio and Latency on Different GPUs

| GPU | FP16 Compute | Original | N=2 S=1 | N=3 S=1 | N=2 S=2 | N=3 S=2 |
|---|---|---|---|---|---|---|
| NVIDIA RTX A5000 | 117 TFLOPS | 1.0x(5.51s) | 1.8x(3.03s) | 2.3x(2.41s) | 3.0x(1.82s) | 4.0x(1.35s) |
| NVIDIA RTX 3090 | 71 TFLOPS | 1.0x(5.61s) | 1.8x(3.20s) | 2.1x(2.65s) | 2.9x(1.91s) | 3.5x(1.60s) |
| NVIDIA RTX 2080Ti | 54 TFLOPS | 1.0x(8.20s) | 1.7x(4.91s) | 2.0x(4.08s) | 2.8x(2.94s) | 3.5x(2.35s) |

# 5 Efficiency Analysis on Different Devices

As a hardware-friendly and versatile method, our acceleration technique delivers strong performance on a wide range of GPUs. We tested inference speeds on the professional-grade NVIDIA RTX A5000, as well as the consumer-grade NVIDIA RTX 2080 Ti and NVIDIA RTX 3090 GPUs. As shown in Table 8, our method achieved a high acceleration ratio across all three GPUs. Furthermore, our method can be applied as long as the devices have basic communication capabilities.

# 6 Conclusion

In this paper, we propose a new parallel paradigm, *AsyncDiff*, to accelerate diffusion models by leveraging model parallelism across multiple devices. We split the denoising model into several components, each assigned to a different device. We transform the conventional sequential denoising into an asynchronous process by exploiting the high similarity of hidden states between consecutive time steps, enabling each component to compute in parallel. Our method has been comprehensively validated on three image diffusion models (SD 2.1, SD 1.5, SDXL) and two video diffusion models (SVD, AnimateDiff). Extensive experiments demonstrate that our approach significantly accelerates inference with only a marginal impact on generative quality. This work investigates the practical application of model parallelism in diffusion models, establishing a new baseline for future research in distributed diffusion models.

# Acknowledgement

This project is supported by the Ministry of Education, Singapore, under its Academic Research Fund Tier 2 (Award Number: MOE-T2EP20122-0006).

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

❈ In this document, we provide supplementary materials that extend beyond the scope of the main manuscript, constrained by space limitations.

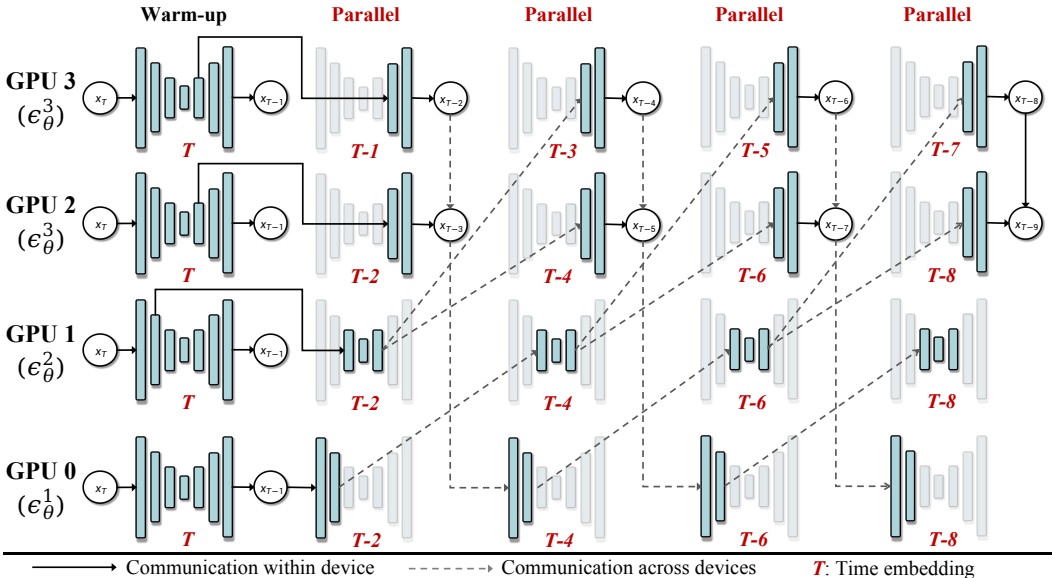

Figure 7: Schematic of the asynchronous diffusion model with stride denoising. The model $\epsilon_\theta$ is divided into three components $\{\epsilon_\theta^n\}_{n=1}^3$, with a stride $S$ of 2 for clarity. A single parallel batch results in the completion of denoising for two steps

## A  More Implementation Details.

**Model Segmentation.** In our method, we partition the cumbersome denoising model into multiple components, each assigned to a different device. After successfully parallelizing the computation of each component, the time cost for each time step now corresponds to the maximum latency among these components. To optimize parallel processing efficiency, we partition the model into segments that each carry a roughly equal computational load. This arrangement allows all modules to finish their computations nearly simultaneously, making full use of available computational resources. The segmentation strategy is sequential except for SDXL [41]. For the denoising U-net within the SDXL module, we group its first and last blocks into a single segment and apply sequential splitting to the remaining blocks. This is because SDXL has specific needs for high-frequency details, and res connections typically contain abundant high-frequency information.

**Time Shifting.** We introduce a technique called time shifting. Following the warm-up steps, the time embedding for each step is shifted back by one step. For instance, in a 50-step asynchronous denoising process with a warm-up of 2 steps, the original sequence of time embeddings is $\{50, 49, 48, 47, ..., 3, 2, 1\}$. With time shifting, this sequence is adjusted to $\{50, 49, 49, 48, ..., 3, 2\}$. In certain extreme cases, asynchronous denoising might leave residual noise in the output. Time shifting addresses this by adjusting the time embeddings backward, enhancing the denoising effect. It's important to note that time shifting is not a standard component of our method but is employed optionally. The quantitative results presented in this paper are achieved without the use of time shifting.

**Stride Denoising.** To further enhance efficiency, we introduce stride denoising, which completes multiple denoising steps simultaneously through a single parallel computation. Figure 7 illustrates the full schematic of applying stride denoising to *AsyncDiff*. In this depiction, the denoising model $\epsilon_\theta$ is divided into three components $\epsilon_\theta^n{}_{n=1}^3$, and for clarity, the stride $S$ is set to 2. Unlike the continuous broadcasting of hidden states at each time step, stride denoising broadcasts them every two steps. As depicted, at time step $\{T-1, T-3, T-5, T-7\}$, we conduct denoising alone, and at time step $\{T-2, T-4, T-6, T-8\}$, we compute and broadcast the hidden states for the next parallel computation round. Consequently, the hidden states from time step $\{T-1, T-3, T-5, T-7\}$

are not required, allowing us to skip the calculations for $\epsilon_\theta^1$ and $\epsilon_\theta^2$ at these steps. Stride denoising effectively reduces both computational load and communication demands by decreasing the parallel computing rounds needed to complete the process. Compared to the significant improvements it brings in efficiency, the quality sacrifice is minimal and can be entirely compensated for by slightly increasing the warm-up steps.

# B   More Analysis.

**Time cost.** In Table 9, we present the time costs associated with model running and inter-device communication when using *AsyncDiff* on SD 2.1. Generally, communication expenses constitute only a minor fraction of the total time cost, demonstrating that *AsyncDiff* is an effective distributed acceleration technique suitable for practical application. It is important to note that as the number of devices increases, the time needed for data broadcasting between devices also rises, thereby increasing the proportion of communication costs. However, employing stride denoising can substantially reduce these costs by decreasing the number of parallel rounds needed to complete the denoising process.

Table 9: Time cost comparisons on SD 2.1. 'Ratio' in this table represents the proportion of communication cost to overall latency. All measurements were conducted on NVIDIA A5000 GPUs equipped with NVLINK Bridge

| Config | Time Cost | | | |
|---|---|---|---|---|
| | Overall | Running | Comm. | Ratio |
| N=2 S=1 | 3.03s | 2.90s | 0.13s | 4.30% |
| N=3 S=1 | 2.41s | 2.18s | 0.23s | 9.54% |
| N=4 S=1 | 2.10s | 1.80s | 0.30s | 14.29% |
| N=2 S=2 | 1.82s | 1.70s | 0.12s | 6.59% |
| N=3 S=2 | 1.35s | 1.25s | 0.10s | 7.40% |

**Speedup Ratio.** We also evaluate the acceleration ratio on SD 2.1 with varying numbers of denoising steps. As indicated in Table 10, *AsyncDiff* significantly enhances processing speed, even with a denoising procedure consisting of only 25 steps. When the number of steps extends to 100, our approach achieves a speedup of up to 4.3x, surpassing the ratio of devices employed.

Table 10: Acceleration ratio on SD 2.1 under different num of denoising steps

| Config | Speedup↑ | | |
|---|---|---|---|
| | 25steps | 50steps | 100steps |
| Origin | 1.0x (2.89s) | 1.0x (5.51s) | 1.0x (10.96s) |
| N=2 S=1 | 1.7x (1.70s) | 1.8x (3.03s) | 1.8x (6.04s) |
| N=3 S=1 | 2.1x (1.35s) | 2.3x (2.41s) | 2.3x (4.71s) |
| N=4 S=1 | 2.4x (1.21s) | 2.6x (2.10s) | 2.7x (4.01s) |
| N=2 S=2 | 2.7x (1.05s) | 3.0x (1.82s) | 3.2x (3.39s) |
| N=3 S=2 | 3.4x (0.86s) | 4.0x (1.35s) | 4.3x (2.52s) |

# C   More Quantitative Results.

To thoroughly assess the quality of images produced following acceleration, we provide quantitative analyses on three base models (SD 2.1 [43], SD 1.5 [43], SDXL [41]) using four additional metrics: the full reference metric, DISTS [5], and no-reference metrics including MUSIQ [20], CLIP-IQA [59], and NIQE [36]. The experimental results in Table 11 demonstrate that our method significantly reduces inference latency while maintaining a high level of quality in diffusion model-generated images. On SD 1.5, our approach not only accelerates the inference process but also brings the image quality closer to the natural distribution.

Table 11: Quantitative evaluations of *AsyncDiff* on three text-to-image diffusion models using more metrics including DISTS [5], MUSIQ [20], CLIP-IQA [59], and NIQE [36].

| Base Model | Configuration | Devices | DISTS↓ | MUSIQ↑ | CLIP-IQA↑ | NIQE↓ |
|---|---|---|---|---|---|---|
| SD 2.1 | Original Model | 1 | – | 69.95 | 0.6653 | 3.9675 |
| | + **Ours** (N=2 S=1) | 2 | 0.1041 | 69.55 | 0.6539 | 3.8850 |
| | + **Ours** (N=3 S=1) | 3 | 0.1280 | 69.04 | 0.6441 | 3.9438 |
| | + **Ours** (N=4 S=1) | 4 | 0.1419 | 68.58 | 0.6365 | 3.9724 |
| | + **Ours** (N=2 S=2) | 3 | 0.1556 | 68.03 | 0.6158 | 3.5761 |
| | + **Ours** (N=3 S=2) | 4 | 0.1689 | 67.13 | 0.5986 | 3.6761 |
| SD 1.5 | Original Model | 1 | – | 71.98 | 0.6534 | 3.5517 |
| | + **Ours** (N=2 S=1) | 2 | 0.1169 | 72.21 | 0.6569 | 3.7448 |
| | + **Ours** (N=3 S=1) | 3 | 0.1434 | 71.73 | 0.6481 | 3.8023 |
| | + **Ours** (N=4 S=1) | 4 | 0.1599 | 71.51 | 0.6442 | 3.8620 |
| | + **Ours** (N=2 S=2) | 3 | 0.1668 | 71.14 | 0.6323 | 3.9613 |
| | + **Ours** (N=3 S=2) | 4 | 0.1905 | 69.42 | 0.6070 | 4.1047 |
| SDXL | Original Model | 1 | – | 71.58 | 0.6633 | 4.0743 |
| | + **Ours** (N=2 S=1) | 2 | 0.1038 | 70.56 | 0.6498 | 4.1139 |
| | + **Ours** (N=3 S=1) | 3 | 0.1211 | 69.88 | 0.6389 | 4.1585 |
| | + **Ours** (N=4 S=1) | 4 | 0.1391 | 67.70 | 0.6056 | 4.0927 |
| | + **Ours** (N=2 S=2) | 3 | 0.1329 | 69.56 | 0.6222 | 4.1685 |
| | + **Ours** (N=3 S=2) | 4 | 0.1527 | 68.16 | 0.5955 | 4.2745 |

## D  More Qualitative Results

**Qualitative Results on Image Diffusion Models.** As depicted in Figure 8, we present further qualitative results for SD 2.1 and SDXL under various configurations. The speedup achieved is nearly proportional to the number of devices utilized, indicating efficient resource usage by our method. Moreover, the images generated by our approach closely match the text descriptions and are of high quality.

**Qualitative Results on Video Diffusion Models.** We present qualitative evaluations of *AsyncDiff* applied to the video diffusion models. Figures 9, 10, and 11 illustrate the generated results using our method on the text-to-video model AnimateDiff [9]. Figure 12 displays results from applying our method to the image-to-video model SVD [2]. For a 50-step video diffusion model, *AsyncDiff* markedly decreases latency—saving tens or even hundreds of seconds—while maintaining the integrity and quality of the generated videos.

## E  Limitations

As a distributed acceleration framework, *AsyncDiff* necessitates frequent communication between devices throughout the denoising process. Consequently, if the devices lack the capability to communicate effectively or have subpar communication infrastructure, our method may not perform optimally. Additionally, *AsyncDiff* operates as a plug-and-play acceleration solution that depends on pre-trained diffusion models. Therefore, if the baseline quality of the original diffusion models is unsatisfactory, achieving high-quality results with our method could be challenging.

## F  Societal impacts

In this paper, we introduce a universal distributed acceleration approach for diffusion models. This method substantially speeds up the inference phase of diverse diffusion models by fully leveraging computational resources. It holds significant potential for practical applications, particularly in computationally intensive generation tasks like video and speech generation.

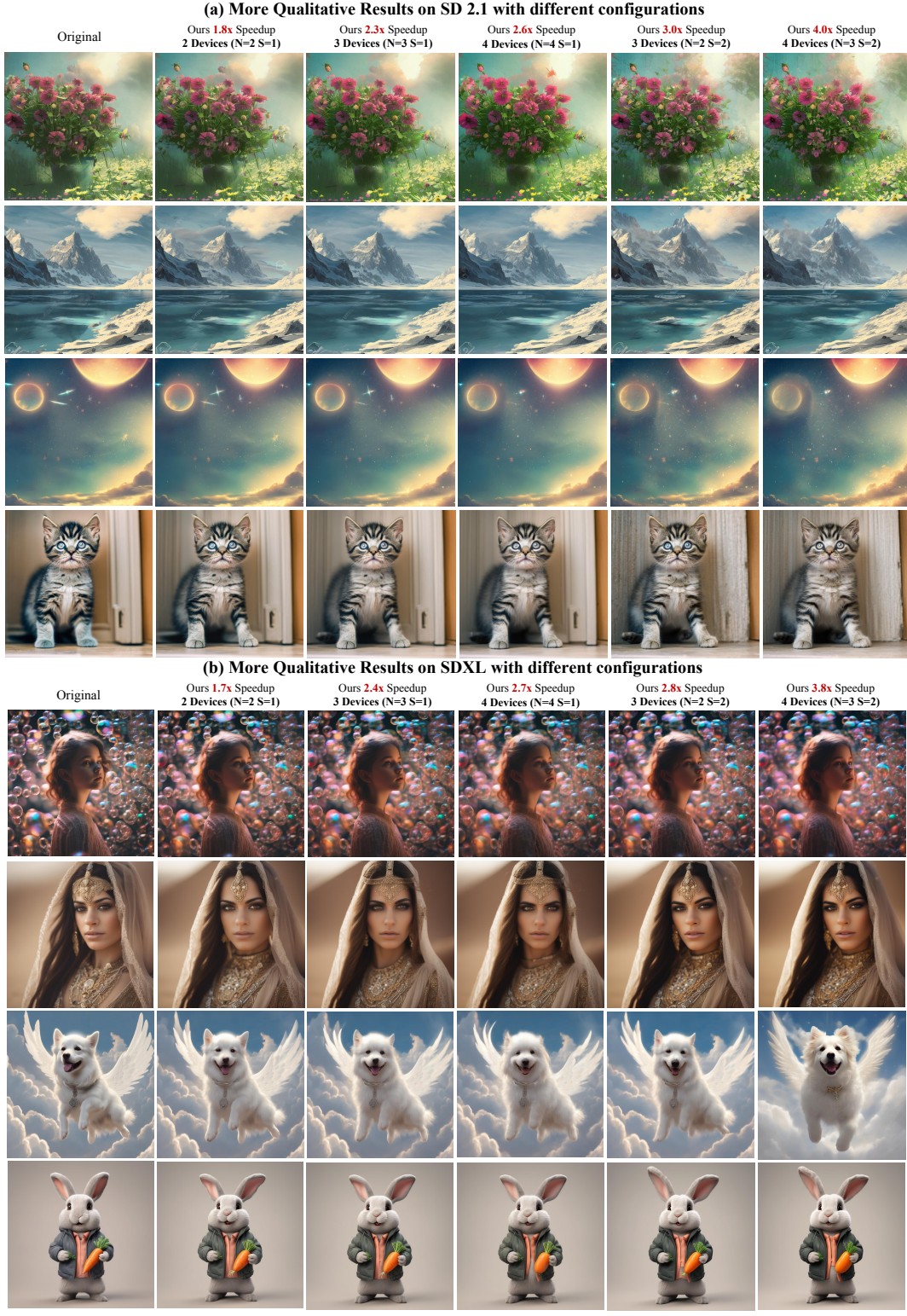

Figure 8: Qualitative results on SD 2.1 and SDXL with different configurations.Our method maintains excellent generation quality even when achieving speedups of up to four times.

*Prompt: Brilliant fireworks on the town, Van Gogh style, digital artwork, illustrative, painterly, matte painting, highly detailed, cinematic*

**Original  43.5s**

**Ours 23.5s  (2 devices)**

**Ours 11.5s  (4 devices)**

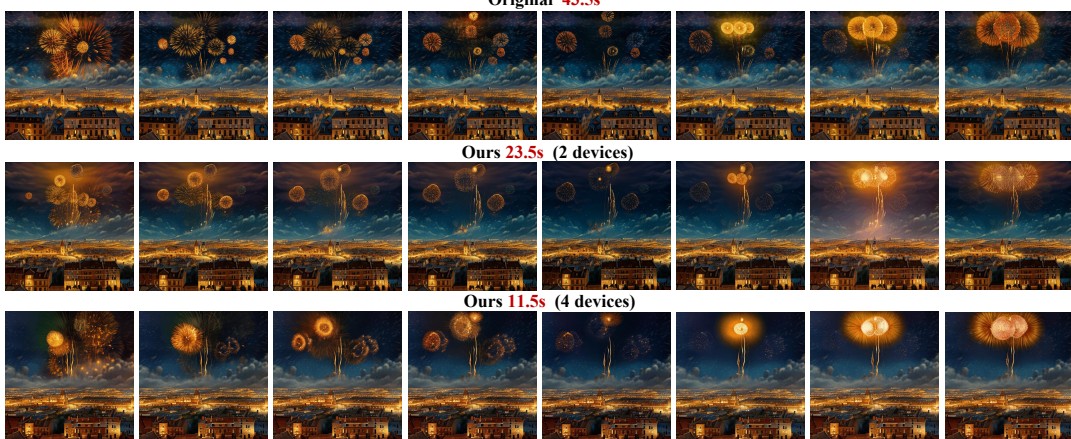

Figure 9: Qualitative results on AnimateDiff (1)

*Prompt: panda playing a guitar, on a boat, in the blue ocean, high quality*

**Original  43.5s**

**Ours 23.5s  (2 devices)**

**Ours 11.5s  (4 devices)**

Figure 10: Qualitative results on AnimateDiff (2)

*Prompt: comic book style, Batman is walking, colored, dynamic background, full body view, clean sharp focus*

**Original  43.5s**

**Ours 23.5s  (2 devices)**

**Ours 11.5s  (4 devices)**

Figure 11: Qualitative results on AnimateDiff (3)

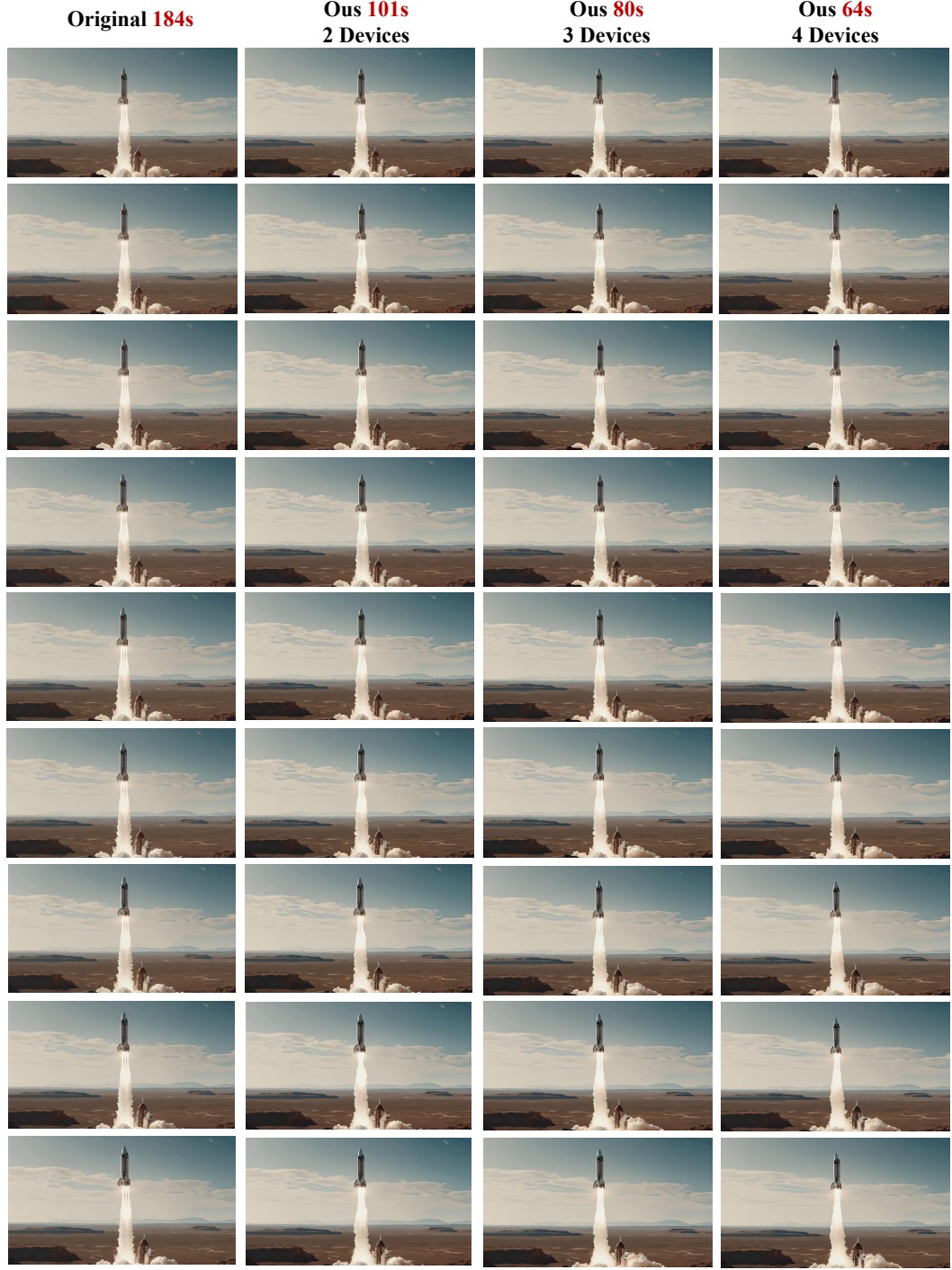

Figure 12: Qualitative results on Stable Video Diffusion

