# OpenReview forum: "AsyncDiff: Parallelizing Diffusion Models by Asynchronous Denoising"
_NeurIPS.cc/2024/Conference — NeurIPS 2024 poster_

### Official Review · Reviewer_773c · 2024-07-09

**Soundness:** 3
**Presentation:** 4
**Contribution:** 3
**Rating:** 6
**Confidence:** 4

**Summary:**

This paper introduces AsyncDiff, an acceleration framework for diffusion models that transforms the traditional sequential denoising process into an asynchronous process. The key insight is that hidden state features in consecutive sampling steps exhibit high similarity. Therefore, feeding the output of the preceding component at time step t-1 as an approximation of the original input to each U-Net component has a negligible impact on performance. This approach allows the parallel denoising processes of AsyncDiff to operate fully asynchronously. Experiments on various versions of Stable Diffusion demonstrate that AsyncDiff significantly speeds up the denoising process for text-to-image generation. Additionally, AsyncDiff is applicable to video generation models such as AnimateDiff and SVD, further showcasing its versatility.

**Strengths:**

1. The writing is overall clear and well-structured. The paper effectively explains the limitations of previous methods that use patch parallelism and introduces a novel approach to "asynchronous" denoising to address these issues.
2. The paper provides thorough comparisons with baseline methods, such as Distrifusion, demonstrating clear improvements in both generation quality and efficiency.
3. The versatility of AsyncDiff is evidenced by experiments conducted on different versions of Stable Diffusion as well as video generation models.
4. The authors address concerns regarding the overhead of communication costs across multiple GPUs, showing that these costs are significantly lower than the model execution time.

**Weaknesses:**

1. Although the main contribution relies on the observation that the hidden states of consecutive steps are similar, the analysis of this phenomenon lacks details. Several key aspects need clearer explanation:
(a) Can the similarity of the hidden states be quantitatively measured? For instance, does a low MSE between hidden states indicate that the two states are “similar”?
(b) Is this phenomenon specific to the U-Net architecture, or is it agnostic to the backbone of the denoising model (e.g., Diffusion Transformer)?
(c) Is it only applicable to DDIM sampling? Does the phenomenon also hold for other fast ODE solvers, such as the DPM solver [1]?

2. While the experiments were conducted using 50 DDIM steps, a comparison with the original model using a number of DDIM steps that achieve a similar speedup would strengthen the argument for AsyncDiff. For instance, in a setup where AsyncDiff achieves a 2.7x speedup, comparing the FID score with the original model using 50 / 2.7 = 19 DDIM steps would clearly demonstrate the necessity of parallelizing the diffusion model.

[1] DPM-Solver: A Fast ODE Solver for Diffusion Probabilistic Model Sampling in Around 10 Steps, Lu et al., NeurIPS 2022

**Questions:**

Most questions are included in the “Weaknesses” section.

Would a similar asynchronous denoising approach be applicable in a conditional generation setup as well? For instance, achieving faster inference for diverse conditional image generation models, such as ControlNet [1] and Zero-1-to-3 [2], could be practically useful.

[1] Adding Conditional Control to Text-to-Image Diffusion Models, Zhang et al., ICCV 2023

[2] Zero-1-to-3: Zero-shot One Image to 3D Object, Liu et al., ICCV 2023

**Limitations:**

The authors discuss the limitations of AsyncDiff (communication cost, dependency on the base model) and the potential societal impacts.

---

> ### Author Rebuttal · Authors · 2024-08-06
>
> ## **Q1: Can the similarity of the hidden states be quantitatively measured? For instance, does a low MSE between hidden states indicate that the two states are “similar”?**
> Thanks for the valuable comment. We provide quantitative analysis and visualization of the hidden state similarities in **Figure 2** and **Figure 3** of the **attached PDF FILE**.
>
> **Figure 2** presents the cosine similarity and MSE of each block's hidden state between adjacent steps, providing a quantitative measure of similarity. The results show that hidden states are highly similar between most steps, except at the initial stage. This high similarity allows for using features from the previous step to parallelize calculations. **Figure 3** visually illustrates this phenomenon by displaying the hidden states of adjacent steps during the diffusion process. Both quantitative and qualitative analyses support our key insight. The initial instability in the curve explains why slightly increasing the warm-up steps can make the AsyncDiff output more consistent with the original output, as shown in Table 2 of our submission.
>
> ## **Q2: Is this phenomenon specific to the U-Net architecture, or is it agnostic to the backbone of the denoising model (Diffusion Transformer)?**
> Thanks for the comment. As a universal acceleration method, AsyncDiff is agnostic to the backbone of the denoising model and is compatible with DiT-based diffusion models. As shown in Table 1, our method significantly speeds up the DiT-Based Stable-Diffusion-3-medium model while maintaining high-quality outputs. The qualitative results in **Figure 1 (b)** of the **attached PDF file** demonstrate that AsyncDiff preserves the excellent image quality and accurate text generation of SD 3.
>
> ### Table 1: Quantitative evaluations of AsyncDiff on DiT-Based SD 3
> *Note: We compare the generative quality when using the DDIM sampler and AsyncDiff to achieve the same speedup. The SD 3 using a default 28-step DDIM sampler is the baseline.*
> | **Method**           | **Speed up ↑** | **Latency ↓** | **CLIP Score ↑** | **FID ↓** |
> |----------------------|----------------|------------|------------------|-----------|
> | SD 3 Original        | 1.0x     | 10.99s        | 32.26           | 33.99     |
> | SD 3 DDIM=15 steps   | **1.8x**     |  6.10s       |  31.90           | 34.55     |
> | **AsyncDiff (N=2 S=1)**| **1.8x**   |  **6.05s**   |  **32.14**       | **32.28** |
>
> ## **Q3: Is it only applicable to DDIM sampling? Does the phenomenon also hold for other fast ODE solvers, such as the DPM solver?**
> Thanks for the valuable feedback. AsyncDiff is a universal method that can be used with various samplers, including the DPM-Solver. Table 2 below provides the quantitative evaluation of AsyncDiff on the SD 2.1 with DPM-Solver sampler. With the same speedup ratio, AsyncDiff significantly improves generation quality compared to the baseline. We also provide the qualitative results in **Figure 1 (a)** of our **attached PDF File**. Our method achieves significant acceleration while maintaining higher consistency with the original output.
>
> ### Table 2: Quantitative evaluations of AsyncDiff using DPM-Solver sampler
> *Note: We compared the generative quality when using the DPM-Solver sampler and AsyncDiff to achieve the same speedup. The SD 2.1 using a 25-step DPM-Solve sampler is the baseline.*
>
> | **Method**           | **Speed up ↑** | **MACs ↓** | **CLIP Score ↑** | **FID ↓** |
> |----------|---------|------------|----------|-----------|
> | SD 2.1 DPM-Solver 25steps  | 1.0x     | 76T        | 31.57            | 28.37     |
> | SD 2.1 DPM-Solver 15steps  | 1.6x     |  46T       |  31.52           | 28.89     |
> | **Our AsyncDiff (N=2 S=1)**| 1.6x     |  **38T**   |  **31.58**       | **27.71** |
> | SD 2.1 DPM-Solver 10steps  | 2.2x     | 30T        |  31.29           | 29.28     |
> | **Our AsyncDiff (N=3 S=1)**| 2.2x     | **25T**    | **31.36**        | **28.20** |
>
> ## **Q4: A comparison with the original model using a number of DDIM steps that achieve a similar speedup would strengthen the argument for AsyncDiff.**
> Thanks for the valuable suggestions. In Table 3, we present a more comprehensive quantitative evaluation of AsyncDiff. Our method achieves significantly better generation quality at similar speeds, and the advantage increases with higher speedup.
>
> ### Table 3: Additional Quantitative Evaluations of AsyncDiff
> *Note: We compared the generative quality when using the DDIM sampler and AsyncDiff to achieve the same speedup. The default 50-step SD 2.1 is the baseline.*
> | **Method**           | **Speed up ↑** | **MACs ↓** | **CLIP Score ↑** | **FID ↓** |
> |-----------|---------|----------|-----------|-----------|
> | SD 2.1 Original      | 1.0x           | 76T        | 31.60            | 27.89     |
> | SD 2.1 DDIM 27steps  | 1.8x           |  41T       |  31.53           | 28.43     |
> | **Our AsyncDiff (N=2 S=1)**| 1.8x     |  **38T**   |  **31.59**       | **27.79** |
> | SD 2.1 DDIM 21steps  | 2.3x           |  32T       |  31.46           | 29.09     |
> | **Our AsyncDiff (N=3 S=1)**| 2.3x     | **25T**    | **31.56**        | **28.00** |
> | SD 2.1 DDIM 15steps  | 3.0x           |  23T       |  31.26           | 30.12     |
> | **Our AsyncDiff (N=2 S=2)**| 3.0x     |  **19T**   |  **31.43**       | **28.55** |
> | SD 2.1 DDIM 11steps  | 4.0x           |  17T       |  30.99           | 32.25     |
> | **Our AsyncDiff (N=3 S=2)**| 4.0x     | **13T**    | **31.22**        | **29.41** |
>
> ## **Q5: Would a similar asynchronous denoising approach be applicable in a conditional generation setup as well?**
> Thanks for the valuable comments. AsyncDiff is a universal method that performs well with conditional generation setups. In **Figure 1 (c)** of the **attached PDF file**, we show how AsyncDiff accelerates ControlNet+SDXL. Our method reduces latency from 19.90s to 14.30s using two A5000 GPUs while nearly mirroring the original output.

---

> > ### Comment · Reviewer_773c · 2024-08-10
> >
> > After thoroughly reviewing the authors' rebuttal, I am satisfied that my original concerns have been addressed. I particularly appreciate the additional experiments with Diffusion Transformers (SD3), and the clear evidence supporting the similarity of the hidden states.
> >
> > In light of this, I have decided to raise my rating to "Weak Accept."

---

> > > ### Author Response · Authors · 2024-08-10
> > >
> > > Thanks for your valuable feedback and time in reviewing my work, your insights are greatly appreciated.

---

### Official Review · Reviewer_7eQm · 2024-07-10

**Soundness:** 4
**Presentation:** 3
**Contribution:** 4
**Rating:** 7
**Confidence:** 4

**Summary:**

This paper proposes AsyncDiff, a plug-and-play acceleration scheme that enables model parallelism across multiple devices. The core method involves dividing the diffusion model into multiple components and executing the inference in parallel. This is facilitated by the high similarity between hidden states in consecutive diffusion steps. With AsyncDiff, it is claimed that off-the-shelf diffusion models can be effectively accelerated with negligible degradation. Therefore, the key contribution of this paper lies in providing a simple and effective method to accelerate the diffusion process through parallelism.

**Strengths:**

1.	The proposed method is well-motivated, and the effectiveness of AsyncDiff has been evaluated on multiple models, making the results compelling.
2.	This method should be useful for video generation, which can take minutes or hours to produce a single video. The paper also supports this with experiments conducted on Stable Video Diffusion.
3.	As shown in Table 5, it seems that the communication cost can be covered by the inference cost, which is a favorable feature for model acceleration.

**Weaknesses:**

1.	Some concepts, such as the “dependency chain” in ABS, are not well explained. It would be beneficial if the author could provide a minimal explanation for these concepts.
2.	The paper appears to share a related idea with Distrifusion [1]. Could the author provide a more detailed discussion about the key differences between them? For example, the main difference seems to be the splitting schemes, where Distrifusion splits the data, and this approach splits the model. What is the main advantage of AsyncDiff?
3.	Are there any failure cases for the proposed AsyncDiff?
[1] DistriFusion: Distributed Parallel Inference for High-Resolution Diffusion Models

**Questions:**

Additional Questions:
1.	Is this method helpful for large batch sizes?
2.	Is this method applicable to more GPUs?

**Limitations:**

I do not find any potential negative societal impact of this work.

---

> ### Author Rebuttal · Authors · 2024-08-06
>
> ## **Q1: Some concepts, such as the “dependency chain” in ABS, are not well explained. It would be beneficial if the author could provide a minimal explanation for these concepts**
> Thank you for the valuable feedback. This is indeed a question that requires further explanation. In the diffusion process, there are two "dependency chains": one between time steps and the other between model blocks.
>
> **Dependency between time steps:** The denoising process is sequential, meaning that denoising at timestep *t* depends on the completion of denoising at timestep *t-1*. This sequential dependency causes latency accumulation and limits the ability to perform parallel inference.
>
> **Dependency between blocks:** The large denoising model is split into *N* components, each with similar computational loads. The input to the *n*th block depends on the output of the *n-1* th block, which poses challenges for running the entire denoising model in parallel.
>
> Our AsyncDiff seeks to disrupt both the dependency chain between time steps and the dependency chain between blocks by leveraging the hidden state's similarity between adjacent steps, allowing for the parallel execution of the denoising process while closely approximating the results of the sequential process.
>
> ## **Q2: Could the author provide a more detailed discussion about the key differences between AsyncDiff and Distrifusion? What is the main advantage of AsyncDiff?**
> Thank you for the comment. As you mentioned, the main difference between AsyncDiff and Distrifusion lies in their core approaches. AsyncDiff focuses on model parallelism, enabling the denoising model to run in parallel regardless of the input data. In contrast, Distrifusion emphasizes data parallelism, parallelizing the computation across different patches of the image.
> Based on these differences, the advantages of AsyncDiff can be summarized as follows:
>
> **1. Wider Applicability:**
> AsyncDiff is a more universal acceleration framework because it focuses on model parallelism, independent of the type of input data or the task of the original model. This makes it suitable for various diffusion-based generation tasks, including text-to-image, text-to-video, image-to-video, super-resolution, ControlNet, speech generation, etc. It can also be used with both Unet-based and DiT-based diffusion models.
>
> **2. Lower Memory Requirements:**
> Distrifusion requires caching activation maps for each layer, significantly increasing GPU memory demands and complicating practical applications. In contrast, AsyncDiff only needs to cache a few hidden states on each device, keeping memory requirements nearly the same as the original model.
>
> **3. Higher GPU Utilization:**
> When the generated image resolution is not very high, Distrifusion can lead to lower GPU utilization, resulting in underutilization of computing resources. AsyncDiff mitigates this issue by not altering the size of the input data, thus maintaining efficient GPU usage. The experimental results in Table 3 of our submission show that AsyncDiff can achieve the same speedup with four GPUs as Distrifusion does with eight GPUs, while also delivering better generation quality.
>
> ### Table 1: Quantitative comparison Distrifusion on SD 2.1
> *To ensure a fair comparison with Distrifusion, we increased the warm-up steps in our method to match the speedup ratio of Distrifusion, allowing us to fairly compare generation quality and resource costs.*
> | **Method**| **Speed up ↑** | **Devices** | **MACs ↓** | **Memory ↓** | **CLIP Score ↑** | **FID ↓** | **LPIPS ↓** |
> |-----|-------|----|---|----|-----|-----|-----|
> | Original Model | 1.0x | 1 | 76T | 5240MB| 31.60| 27.87 | --|
> | **Distrifusion** | 1.6x | **2** | **38T**| 6538MB | **31.59**| 27.89| **0.0178**  |
> | **Ours (N=2 S=1)** | 1.6x | **2** | 44T    | **5450MB** | **31.59**| **27.79** | 0.0944 |
> | **Distrifusion**| 2.3x | 4     | **19T**| 7086MB | 31.43| 27.97 | 0.2710 |
> | **Ours (N=2 S=2)** | 2.3x | **3** | 20T    | **5516MB**| **31.49**| **27.71** | **0.2117**  |
> | **Distrifusion**| 2.7x | 8| **10T**| 7280MB| 31.31 | 28.12     | 0.2934 |
> | **Ours (N=3 S=2)**| 2.7x | **4** |14T| **5580MB**| **31.40**| **28.03** | **0.1940**  |
>
> ## **Q3: Are there any failure cases for the proposed AsyncDiff?**
> As an acceleration method, AsyncDiff depends on the quality of the original base model. If the base model's generation quality is not good, AsyncDiff will also produce unsatisfactory results. While our method can significantly speed up the process and maintain the original generation quality, it typically cannot enhance the base model's original generation quality.
>
> ## **Q4:Is this method helpful for large batch sizes?**
> Thanks for the comment. As shown in the following Table 2, AsyncDiff will still be helpful for large batch sizes.
> ### Table 2: Acceleration and Latency under Different Batchsizes
> *We evaluate the running speed of AsyncDiff on SD 2.1. The latency is evaluated using Nvidia RTX A5000 GPUs, CUDA 12.0, and PyTorch 2.1.*
> | BatchSize|Original|N=2 S=1|N=4 S=1|N=3 S=1|N=2 S=2|N=3 S=2|
> |----|-----|-----|-----|-----|-----|-----|
> | 1| 1.0x(5.51s)| 1.8x(3.03s)  | 2.3x(2.41s)| 2.6x(2.10s)| 3.0x(1.82s)| 4.0x(1.35s)|
> | 4| 1.0x(19.34s)| 1.7x(11.46s)| 2.2x(8.60s)| 2.5x(7.85s) | 2.9x(6.75s)| 4.1x(4.77s)|
> | 8| 1.0x(36.19s)| 1.7x(21.46s)| 2.2x(16.44s)| 2.4x(14.90s) | 2.8x(12.75s)| 4.0x(9.06s)|
>
> ## **Q5:Is this method applicable to more GPUs?**
> Thanks for the comment. Asyncdiff can be applicable to more GPUs. However, we do not recommend splitting the model into more than four components, as the speed gains are minimal. In practice, combining AsyncDiff with other data parallel methods is more effective.  Asyncdiff is compatible with any other data-parallel solution (batch splitting, patch splitting). Taking advantage of model parallelism and data parallelism at the same time will achieve a more extreme speedup ratio while minimizing the sacrifice of generation quality.

---

### Official Review · Reviewer_iHdZ · 2024-07-10

**Soundness:** 3
**Presentation:** 4
**Contribution:** 4
**Rating:** 7
**Confidence:** 5

**Summary:**

This work introduces an acceleration method for diffusion models by distributing the model blocks across multiple GPUs and running different blocks asynchronously. The core motivation of this paper lies in the observation that the hidden states of a block exhibit high similarity across consecutive diffusion steps. This enables the execution of different blocks based on pre-computed results from earlier steps. The proposed method was evaluated on Stable Diffusion v2.1, v1.5, and XL, achieving a 1.8x to 4x acceleration with minimal performance loss on CLIP score and FID.

**Strengths:**

1) The idea of parallelizing the inference of different blocks is quite interesting. The method is general and can be applied to several diffusion models like Stable Diffusion, Stable Video Diffusion, and AnimateDiff. The method shows robustness to different speed-up ratios as shown in Tables 3, 4, and 5.

2) Extensive experiments were conducted in this paper, encompassing both image generation and video generation. And the results are good: the proposed method as able to achieve significant acceleration with slight performance degradation.

**Weaknesses:**

1) Is the communication cost huge compared to the inference cost? As shown in Table 3, the number of GPUs used in this paper is primarily 2 or 3. Will there be issues if this is extended to 8 or 16 GPUs? Is this method still competitive when the number of devices exceeds 8, where cross-node communication becomes substantial?

2) The experimental section mainly focuses on quality metrics such as FID and CLIP Score. It would be beneficial if the authors could provide more analytical results about the method, such as the similarity of hidden states.

3) Figure 5: “Quantitative” should be “Qualitative”.

**Questions:**

1) What happens if the total number of timesteps is not divisible by the number of GPUs? Is the proposed method still applicable in this scenario?

2) Is the proposed method still effective on low-end GPUs or edge devices?

**Limitations:**

N/A.

---

> ### Author Rebuttal · Authors · 2024-08-06
>
> ## **Q1: Is the communication cost huge compared to the inference cost? Will there be issues if this is extended to 8 or 16 GPUs? Is this method still competitive when the number of devices exceeds 8, where cross-node communication becomes substantial?**
> Thanks for the valuable comment, this is indeed a question that needs further explanation
>
> As demonstrated in Table 1, the communication cost is minimal compared to the inference cost.
>
> Applying to 8 or 16 GPUs can lead to increased communication costs due to GPU architecture. In a common 8-GPU node, GPUs 0, 1, 2, and 3 form one group, while GPUs 4, 5, 6, and 7 form another. Cross-group communication is about 40% more costly than within a group, the cross-node cost is even higher. This is a shared challenge for all distributed inference frameworks, both in the Diffusion and LLM areas.
>
> Our proposed stride denoising technique effectively alleviates this issue by reducing communication frequency. As shown in Table 2, during a 50-step diffusion process, AsyncDiff requires only 25 communications. This significant reduction means our method is less affected by increased cross-group or cross-node communication costs, keeping it competitive. In the future, we will explore more innovative technologies to further address this general challenge.
>
> ### Table 1: Time cost comparisons on SD 2.1
> *Note: 'Overall' represents the overall latency, 'Running' represents the time cost of model running, 'Comm.' represents the communication time cost. 'Ratio' in this table represents the proportion of communication cost to overall latency. All measurements were conducted on NVIDIA A5000 GPUs.*
> | **Configuration** | **Devices** | **Overall** | **Running** | **Comm.** | **Ratio** |
> |------------|-------------|-------------|-------------|-----------|------------|
> | AsyncDiff N=2 S=1    | 2 GPUs| 3.03s       | 2.90s       | 0.13s     | 4.30%     |
> | AsyncDiff N=3 S=1    | 3 GPUs| 2.41s       | 2.18s       | 0.23s     | 9.54%     |
> | AsyncDiff N=4 S=1    | 4 GPUs| 2.10s       | 1.80s       | 0.30s     | 14.29%    |
> | AsyncDiff N=2 S=2    | 3 GPUs| 1.82s       | 1.70s       | 0.12s     | 6.59%     |
> | AsyncDiff N=3 S=2    | 4 GPUs| 1.35s       | 1.25s       | 0.10s     | 7.40%     |
>
> ### Table 2: Effect of stride denoising in the 50-step Diffusion Process
> *Note: Stride denoising significantly lowers communication costs by decreasing the communication frequency*
>
> | **Configuration** |Communication Nums↓| Communication Freq↓|
> |------------|-------------|-------------|
> | *AsyncDiff* w/o stride denoising  | 49 times | 0.98/timestep|
> | *AsyncDiff* w/ stride denoising  |  **25 times** | **0.50/timestep**|
>
> ## **Q2:  It would be beneficial if the authors could provide more analytical results about the method, such as the similarity of hidden states.**
> Thank you for your valuable comment. We provide quantitative analysis and visualization of the hidden state similarities in **Figure 2** and **Figure 3** of the **attached PDF FILE**.
>
> **Figure 2** shows the cosine similarity and MSE for each block's hidden state between adjacent steps. As illustrated, the hidden states are highly similar between most steps, except at the initial stage. This high similarity enables us to use features from the previous step to parallelize calculations. **Figure 3** further visualizes this phenomenon by displaying the hidden states of adjacent steps in the diffusion process. The initial instability demonstrates why slightly increasing the warm-up step can make the AsyncDiff output image pixel-wise consistent with the original output, which is also shown in Table 2 in our submission.
>
> ## **Q3: Figure 5: “Quantitative” should be “Qualitative”.**
> Thanks for the careful review. We will correct this error in the next version.
>
> ## **Q4: What happens if the total number of timesteps is not divisible by the number of GPUs? Is the proposed method still applicable in this scenario?**
> Thank you for the valuable comment. In this case, AsyncDiff will not face any issues. Although we split the denoising model into *N* parts, we still obtain one timestep of noise after each parallel batch. However, the runtime for this step in the denoising model is reduced to *1/N*.
>
> This is supported by the experimental results in Table 1 of our submission. When accelerating a 50-step diffusion model, AsyncDiff performs well, even when using 3 or 4 GPUs.
>
> ## **Q5: Is the proposed method still effective on low-end GPUs or edge devices?**
> Thank you for the valuable comment. This is a very practical question.
> We conducted additional experiments to show the acceleration ratio of AsyncDiff on consumer-grade graphics cards (NVIDIA RTX 2080 Ti GPUs). As shown in Table 3, our method still achieved a significant speedup on these GPUs.
>
> ### Table 3: Acceleration Ratio and Latency on low-end GPUs
> *Note:  Acceleration Ratio and Latency are evaluated using CUDA 12.0 and PyTorch 2.1*
>
> | **GPU**     | **FP16 Compute** |Original|N=2 S=1|N=3 S=1|N=2 S=2|N=3 S=2|
> |-------------|------------------|-------------|-------------|-------------|-----------|-----------|
> | NVIDIA RTX 2080Ti | 54  TFLOPS| 1.0x(8.20s)| 1.7x(4.91s)  | 2.0x(4.08s) | 2.8x(2.94s)| 3.5x(2.35s)|

---

> > ### Comment · Reviewer_iHdZ · 2024-08-13
> > **Final Rating**
> >
> > The rebuttal from the author has addressed all my concerns, thus I would like to keep my rating as accept.

---

> > > ### Author Response · Authors · 2024-08-13
> > >
> > > Thank you for the insightful and constructive comment. Your expertise and time are greatly appreciated in helping improve my work's quality.

---

### Official Review · Reviewer_E7zL · 2024-07-13

**Soundness:** 4
**Presentation:** 3
**Contribution:** 3
**Rating:** 6
**Confidence:** 3

**Summary:**

This paper proposes AsyncDiff to enable diffusion model parallelism across multiple devices and achieve a very impressive speedup with negligible degradation. Specifically, the denoising model is split into several components, each assigned to a different device. The conventional sequential denoising process is transformed into an asynchronous one by exploiting the high similarity of hidden states between consecutive time steps, enabling each component to compute in parallel.

**Strengths:**

1. AsyncDiff is a universal and plug-and-play acceleration scheme and can enable model parallelism across multiple devices. The acceleration that AsyncDiff achieves on image and video diffusion models is impressive
2. The motivation is clear and the experiments are comprehensive.

**Weaknesses:**

1. The discussion and comparison with the existing diffusion sampler is not sufficient, whether AsyncDiff can be combined with other diffusion samplers such as DPM-Solver.
2. In the absence of an explanation of the GPU, and AsyncDiff's ablation experiments on different GPU devices, can other GPUs achieve a good acceleration ratio?

**Questions:**

I do not have other questions about this paper.

**Limitations:**

As stated by the author, AsyncDiff necessitates frequent communication between devices throughout the denoising process. Therefore, if the devices lack the capability to communicate effectively, AsyncDiff may not perform optimally.

---

> ### Author Rebuttal · Authors · 2024-08-06
>
> ## **Q1: Whether AsyncDiff can be combined with other diffusion samplers such as DPM-Solver**
> Thanks for the valuable feedback. AsyncDiff is a universal method that can be used with various samplers, including the DPM-Solver. Table 1 below provides the quantitative evaluation of AsyncDiff on the SD 2.1 with DPM-Solver sampler. With the same speedup ratio, AsyncDiff significantly improves generation quality compared to the baseline. We also provide the qualitative results in **Figure 1 (a)** of our **attached PDF File**. Our method achieves significant acceleration while maintaining higher consistency with the original output.
>
> ### Table 1: Quantitative evaluations of AsyncDiff using DPM-Solver sampler
> *Note: We compared the generative quality when using the DPM-Solver sampler and AsyncDiff to achieve the same speedup. The SD 2.1 using a 25-step DPM-Solve sampler is the baseline.*
>
> | **Method**           | **Speed up ↑** | **MACs ↓** | **CLIP Score ↑** | **FID ↓** |
> |----------------------|----------------|------------|------------------|-----------|
> | SD 2.1 DPM-Solver 25steps  | 1.0x     | 76T        | 31.57            | 28.37     |
> | SD 2.1 DPM-Solver 15steps  | 1.6x     |  46T       |  31.52           | 28.89     |
> | **Our AsyncDiff (N=2 S=1)**| 1.6x     |  **38T**   |  **31.58**       | **27.71** |
> | SD 2.1 DPM-Solver 10steps  | 2.2x     | 30T        |  31.29           | 29.28     |
> | **Our AsyncDiff (N=3 S=1)**| 2.2x     | **25T**    | **31.36**        | **28.20** |
>
>
> ## **Q2: Can other GPUs achieve a good acceleration ratio?**
> Thanks for the valuable comments. We conducted additional experiments to compare the acceleration ratio of AsyncDiff on three different GPUs (NVIDIA RTX A5000, NVIDIA RTX 3090, and NVIDIA RTX 2080 Ti). As shown in Table 2, our method achieved a significant speedup across all tested GPUs.
>
> ### Table 2: Acceleration Ratio and Latency on Different GPUs
> *Note:  Acceleration Ratio and Latency are evaluated using CUDA 12.0 and PyTorch 2.1*
>
> | **GPU**     | **FP16 Compute** |Original|N=2 S=1|N=3 S=1|N=2 S=2|N=3 S=2|
> |-------------|------------------|-------------|-------------|-------------|-----------|-----------|
> | NVIDIA RTX A5000   | 117 TFLOPS| 1.0x(5.51s)| 1.8x(3.03s)  | 2.3x(2.41s) | 3.0x(1.82s)| 4.0x(1.35s)|
> | NVIDIA RTX 3090    | 71  TFLOPS| 1.0x(5.61s)| 1.8x(3.20s)  | 2.1x(2.65s) | 2.9x(1.91s)| 3.5x(1.60s)|
> | NVIDIA RTX 2080Ti | 54  TFLOPS| 1.0x(8.20s)| 1.7x(4.91s)  | 2.0x(4.08s) | 2.8x(2.94s)| 3.5x(2.35s)|

---

> > ### Comment · Reviewer_E7zL · 2024-08-08
> >
> > The authors have posted an effective rebuttal that resolves my concerns. I still vote for a weak acceptance of this work.

---

> > > ### Author Response · Authors · 2024-08-08
> > >
> > > We sincerely appreciate your dedicated time and effort in reviewing our submission. Your valuable feedback is greatly appreciated.

---

### Author Response · Authors · 2024-08-06

Dear Reviewers, Chairs,

We sincerely appreciate the time and effort you have spent evaluating our submission, and we look forward to the discussion stage. We will include the review stage results in the appendix of our next version.

---

### Author Rebuttal · Authors · 2024-08-06

Dear Chairs,

We sincerely appreciate the time and effort you have spent evaluating our submission, and we look forward to the discussion stage. We will include the review stage results in the appendix of our next version.

---

### Comment · Area_Chair_71QC · 2024-08-10

Dear reviewers,

Could you please respond to the rebuttal, discuss with authors and finalize your score?

---

### Decision · Program_Chairs · 2024-09-25

**Decision:**

Accept (poster)

**Comment:**

The paper introduces AsyncDiff, a novel framework that accelerates diffusion models by enabling asynchronous parallelism across multiple devices, effectively utilizing the high similarity of hidden states between consecutive steps. The method achieves impressive speedups (1.8x to 4x) on various diffusion models, including image and video generation, with minimal impact on performance metrics like FID and CLIP score. Although some reviewers expressed a desire for more detailed analysis of communication costs and the underlying similarity of hidden states, the proposed method's versatility and significant practical benefits make it a strong contribution to the field. Given its potential impact, I recommend accepting the paper.